# Effect of the Law Enforcement Duty Belt on Muscle Activation during Hip Hinging Movements in Young, Healthy Adults

**DOI:** 10.3390/jfmk8030099

**Published:** 2023-07-17

**Authors:** James W. Kearney, Megan N. Sax van der Weyden, Nelson Cortes, Orlando Fernandes, Joel R. Martin

**Affiliations:** 1Sports Medicine Assessment Research & Testing (SMART) Laboratory, George Mason University, Manassas, VA 20110, USA; jkearne@gmu.edu (J.W.K.); msaxvand@gmu.edu (M.N.S.v.d.W.); 2School of Sport, Rehabilitation, and Exercise Science, University of Essex, Colchester CO4 3WA, UK; n.cortes@essex.ac.uk; 3Department of Bioengineering, George Mason University, Fairfax, VA 22030, USA; 4Sport and Health Department, School of Science and Technology, University of Évora, 7004-516 Évora, Portugal

**Keywords:** lower back, injury, police military, firefighter, comfort, restriction, tactical

## Abstract

Sixty percent of all law enforcement officers (LEOs) experience low back pain (LBP), with the LEO duty belt (LEO_DB_) commonly reported to be a contributing factor. The primary purpose of the study was to investigate the LEODB’s effect on muscular activity and compare it to a tactical vest, which is a commonly used alternative to an LEO_DB_. In total, 24 participants (13 male, 11 female; mass, 73.0 ± 11.1 kg; height, 169.0 ± 10.0 cm; age, 24.0 ± 5.8 years) completed a progressive series of hip hinge tasks in a single testing session. All participants completed four conditions (no belt, leather belt, nylon belt, and weight VEST) in a randomized order. Surface electromyography (sEMG) sensors were placed bilaterally on the rectus abdominus, multifidus, biceps femoris, and rectus femoris. Across all tasks, no significant effects of load on muscle activity were found for any of the muscles. Participants rated the VEST condition as more comfortable (*p* < 0.05) and less restrictive (*p* < 0.05) than either LEODB. The findings suggest an LEO_DB_ does not alter muscle activity during bodyweight hip hinging or lifting objects from the ground. Future research should examine whether changes in muscle activity occur with durations of LEO_DB_ wear more similar to an actual work shift duration for LEOs (≥8 h).

## 1. Introduction

Law enforcement officers experience lower back pain at an equal or greater frequency than the general public [1,2,3,4]. In fact, 60% of all law enforcement officers (LEOs) will experience lower back pain (LBP) [1,4]. LBP in LEOs can result in a decrease in quality of life [5], missing work [5,6], early retirements [5,6,7], and medical disability [4,5,6,7]. These factors alone amount to an annual cost of up to USD 56 billion [8]. LBP in LEOs is attributed to multiple factors, including prolonged sitting, heavy lifting, and wearing a law enforcement duty belt (LEO_DB_) [7]. There is a substantial body of literature reporting the association between heavy lifting and the development of LBP [9,10]. However, there is a lack of peer-reviewed research regarding the effects of an LEO_DB_ on known biomechanical parameters associated with LBP while performing lifting tasks.

The previous literature has reported that the posture of the hips and spine affects the activation patterns of the surrounding musculature due to altered length–tension relationships [11,12]. Specifically, an LEO_DB_ can alter the resting hip position while standing [7], resulting in a significant anterior rotation and increased lumbar lordosis, especially when the belt is loaded anteriorly [6]. As a result, wearing an LEO_DB_ may lead to changes in the muscle activity of the core musculature, which are often seen in individuals with LBP [13]. Altered muscle activity can have substantial effects on internal forces on the spine. For example, McGill and Kippers [14] found that when 8 kg loads were held anteriorly, while in full hip flexion with relaxed spinal extensors, subjects experienced 3 kN more compressive force and 755 N more anterior shear force. Therefore, changes in muscle activity due to an LEO_DB_ may increase internal forces on the spine and be a contributing factor to the high instances of LBP in LEOs [14,15]. While measuring forces within the spine during activity is difficult, muscle activity can be used as a surrogate measure, which can be assessed with surface electromyography (sEMG) [16].

Occupational lifting has been shown to increase the likelihood of experiencing LBP and missing work due to LBP [17,18,19]. Specifically, lifting heavy loads off the floor and dragging heavy loads across the floor were found to be two common mechanisms for new instances of LBP [19]. Many tasks LEOs perform on a regular basis involve hinging at the hips to lift and drag loads [4,8,18,19,20]. One common example of a hip hinge in LEOs is the dummy drag assessment recruits must complete to graduate from the academy, which simulates an occupational task law enforcement officers would need to perform throughout their career [21]. The dummy drag assessment has recruits squat down, wrap their arms around a supine 56–75 kg weighted mannequin, and then drag said mannequin 9–18 m. The dummy drag places a high amount of stress on the spinal erectors like that of traditional hip hinge exercises such as the deadlift [13,22,23,24].

The hip hinge demands a high degree of activity from the spinal erectors and supportive roles for the hamstrings, abdominals, and gluteal muscles [22,23,24,25]. Depending on load and stance, the hip hinge’s activity in the erector spinae from T10-L3 will be 88–98% of MVIC [24] and 24–100% of MVIC in the semimembranosus and semitendinosus [22,23], 60% of MVIC in the rectus abdominus [23], and 35–37% of MVIC in the gluteus maximus [23]. Improper hip hinge form in LEOs might be from several factors such as tissue flexibility [14,15], motor control [4,20,21], chronic injury [25,26,27], or an LEO_DB_ [28,29,30]. Regardless of the cause, when hip hinge movements are performed incorrectly, several negative biomechanical outcomes can occur. Theoretically, if performed incorrectly, the mechanical stress would shift from the supportive musculature to the ligamentous structures of the lower back [14].

Another potential factor that might contribute to LBP in law enforcement officers is the subjective comfort and restriction of the vest and LEO_DB_ [31]. Many law enforcement officers anecdotally report significant discomfort and even pain from wearing an LEO_DB_. Several studies have examined the impact of equipment comfort on officers′ occupational performance [30,32]. In a study by Schram et al. [30], a larger vest with dispersed load was found to be more comfortable and resulted in better shot accuracy. However, it has also been shown that body amour, such as a tactical vest, is restrictive via reduced measures of mobility [33]. Moreover, Ramstrand and colleagues [32] found that both the vest and LEO_DB_ decreased measures of range of motion and altered gait patterns, suggesting both forms of law enforcement load carriage are restrictive. However, it is currently unclear if these restrictions result in altered muscle activity in the core musculature that could subsequently lead to LBP in LEOs.

Despite the high prevalence of LBP in LEOs and anecdotal reports of the LEO_DB_ contributing to LBP, to our knowledge, no studies examining the effects of an LEO_DB_ on muscle activity during hip hinge movements exist. Therefore, the primary aim of this study was to examine the muscle activity during hip hinge movements while wearing an LEO_DB_ as compared to control condition with no LEO_DB_. Additionally, it is common in certain situations for LEOs to wear either a leather LEO_DB_, a nylon LEO_DB_, or a tactical vest instead of an LEO_DB_. Thus, the second aim of the present study was to compare the two types of LEO_DB_ and the tactical vest, when performing the hip hinging tasks. Lastly, the third aim was to assess the variation in perceived comfort and restriction of these three forms of LEO load carriage. Due to the lack of prior evidence regarding the effects of an LEO_DB_ on muscle activity and comfort, particularly during hip hinge movements, the null hypothesis was tested for each aim.

## 2. Materials and Methods

### 2.1. Study Design

This study used a randomized, prospective study design to determine the effect of an LEO_DB_ on muscular activity and feelings of restriction and comfort in a young and healthy population. All testing was performed during a voluntary single 150 min session. All data collection followed a standardized procedure headed by 1–3 members of the research team (JWK, MNS, and JRM). All data collection procedures were in accordance with the Helsinki Declaration [34].

### 2.2. Participants

An a priori power analysis was conducted with G-Power (Version 3.1) statistical software. The analysis was conducted based on previous studies that reported the effects of load on mean muscle activity during activities such as standing [35] and walking [36,37,38]. The parameters entered in the power analysis were a small effect size of 0.3, power (1 − β) of 0.8, and an alpha (α) of 0.05. Power analysis indicated that a minimum of 24 participants would be needed for sufficient statistical power. The inclusion criteria were as follows: participants must be between 18 and 45 years of age, BMI < 30 kg/m^2^, and recreationally active at least 3 days a week. The exclusion criteria were no previous history of lower back or lower extremity injury within the past 6 months, a lack of familiarity with the deadlift exercise (a common hip hinge), or an inability to deadlift one’s body mass via a response to the entry questionnaire. Any participant who stated, verbally or within the questionnaire, that they have had lower back pain within the past 6 months was immediately removed from study participation. A total of 29 eligible participants completed the experimental procedures; however, 5 were removed due to technical issues with sEMG sensors resulting in a total of 24 participants (13 male, 11 female; mass, 73.0 ± 11.1 kg; height, 169.0 ± 10.0 cm; age, 24.0 ± 5.8 years, BMI 25.4 ± 2.4 kg/m^2^). The study was approved by the Institutional Review Board at George Mason University (IRB approval #: 1455213-1). Upon arrival, participants completed an informed consent form and were screened to ensure eligibility requirements were met.

### 2.3. Procedures

The data reported in the present study were part of a larger, ongoing project to investigate the biomechanical effects of wearing an LEO_DB_ on common movement tasks performed by LEOs. Each participant for this project was tested individually. Participants visited the laboratory during a single testing session (Figure 1). Prior to participants arriving at the laboratory, the load condition order was randomly determined via Google’s (GOOGLE, Mountain View, CA, USA) online random number generator (1 = Control, 2 = Leather LEO_DB_, 3 = Nylon LEO_DB_, and 4 = Vest).

Once in the laboratory, participants completed a pretest screening and signed an informed consent form. Afterward, the participants’ anthropometrics and age data were recorded. The participants then completed a series of surveys regarding moods, personality, and lifestyle behaviors (Qualtrics, XM, Provo, UT, USA). The total time to complete the survey instruments was approximately 20 min, after which, participants completed a brief warm-up. This warm-up was only conducted prior to the first round of assessment. Following the warm-up, the participants performed a series of tasks for each of the randomized 4 conditions. Condition 1 consisted of no loaded vest or LEO_DB_ and served as the control. In condition 2, the participants wore a 7.2 kg leather LEO_DB_ with loaded pouches and a holster. In condition 3, the participants wore a 7.2 kg nylon LEO_DB_ with loaded pouches and a holster. In condition 4, the participants wore a 7.2 kg weighted vest to simulate a law enforcement duty vest. The leather and nylon LEO_DB_ of conditions 2 and 3 were identical in terms of load placement and dimensions, with the only difference being the material the belt was made of.

For each condition, a battery of cognitive, static, and dynamic movement tasks was performed, including a hip hinge progression. For the aim of the present study, the hip hinging movements will be the focus of the subsequent sections. While the conditions were randomized, the tasks for each condition were performed in a constant order, with the hip hinge progression performed last. The fixed task order was determined after pilot testing and in consideration with warm-up guidelines from the National Strength and Conditioning Association (NSCA) [39]. Subjective feedback from participants during pilot testing was that the procedures used in the current study were not fatiguing and the intensity of tasks within a condition went from lowest (i.e., standing tasks) to highest intensity (i.e., heaviest loaded hip hinge). Time to complete the study was approximately 150 min, with each condition typically lasting 20–25 min. To avoid circadian rhythm effects on neuromuscular performance, all testing was conducted in the afternoon between 12 pm and 3 pm [40]. All participants were given 5 min of rest between each condition trial and 30 s to 1 min of rest between each task.

#### 2.3.1. Warm-Up

In the field, LEOs are not afforded the opportunity to warm-up in emergency situations. However, the warm-up was included in our procedures to ensure safety of participants and was designed in a manner to also not induce any fatigue prior to experimental trials. Participant completed a warm-up routine that required performing 2 rounds of 4 body weight movements. These movements included 10 bird dogs, 5 inchworms, 12 bodyweight squats, and 12 bodyweight Romanian deadlifts (RDLs). Rest between movements and rounds was set by the participant. 

#### 2.3.2. Hip Hinge Progression

Hip hinge movements consisted of an unloaded hip hinge, 20 kg lift, and 43 kg lift. These loads were selected based on the OSHA lifting guidelines [41] and fitness requirements for LEOs [19]. Approximately 60 s of rest was given between each task and 5 min of rest between conditions.

Participants completed hip hinges to a 21 cm box with no added weight besides that provided by their belt or vest condition. At a pace of 60 bpm, the participant bent at the hips and touched the box with both hands. The instructions were to keep their arms straight and only move at the waist. sEMG activity data were collected for 30 s while participants performed the hinge.

Next, the subjects used a hip hinge movement pattern to lift loads of 20 kg and 43 kg from the floor. An unloaded hexagonal deadlift bar (hexagonal bar; 20 kg) was placed at 21 cm height. For the final lifting trial, 2 weight plates were added to the hexagonal bar to increase the load to 43 kg. As with the 20 kg trial, participants hinged down to pick up the hexagonal bar and performed three complete repetitions. The height of this load matched the height of the unloaded and 20 kg trials, and was kept constant for all subjects. Afterward, the relative load of the condition and each of the loaded hip hinge trials was calculated for each participant [42]. While setting the loads at a constant height means taller participants had a greater range of motion to travel, the depth of this starting position is similar to the position all LEO recruits must maintain during the dummy drag assessment [21]. Repetitions were completed at the participant’s self-selected pace during a 15 s data collection window. Visual depictions of the three hip hinge assessments and a general outline of the methods are shown in Figure 1.

Pilot testing revealed that the rectus femoris sEMG obstructed the straight barbell path and would result in the barbell, forcefully contacting the sEMG sensors. This led to the use of a hexagonal deadlift during the study to avoid this issue. Additionally, during pilot testing, all participants completed 3 repetitions in 8 to 13 s in all cases. Thus, participants were given 15 s to hinge down, pick up the hexagonal barbell, and perform 3 complete repetitions. A repetition was deemed complete when the participant started in the bottom position, lifted the load to a standing position, and returned to the starting position. Repetitions that did not result in an upright trunk and fully extended hips and knees were considered incomplete, resulting in the trial being discarded and repeated.

### 2.4. Measures

#### 2.4.1. Anthropometrics

Participants’ height was measured using a stadiometer (Detecto, Webb City, MO, USA) to the nearest 0.01 cm and mass was measured via a digital scale (EatSmart, Tokyo, Japan) to the nearest 0.1 kg.

#### 2.4.2. Physical Activity and Sedentary Behavior

During the initial screening process, participants completed the International Physical Activity Questionnaire-Short Form (IPAQ-SF), a widely used and reliable tool for assessing physical activity levels [43,44]. The IPAQ-SF consists of seven open-ended items that inquire about participants′ physical activity during the past seven days. These items prompt participants to report the number of days per week, as well as the total duration in hours and minutes, for engaging in vigorous, moderate, and light physical activities. Additionally, participants were asked to provide the number of hours and minutes they spent sitting throughout the week. To calculate the total minutes spent on vigorous, moderate, and light activities, as well as sitting time, the following formula was used: “(number of hours × 60 min) + number of minutes”. To derive a moderate–vigorous physical activity (MVPA) score, the value for vigorous activity was doubled and combined with the time spent on moderate activity [45]. The MVPA score was then used to categorize as active (≥150 min/week) or inactive (<150 min/week) according to current recommended guidelines [46].

#### 2.4.3. Muscle Activity

Surface electromyography (sEMG) sensors (Trigno, Delsys, MA, USA) were placed bilaterally on the multifidus (MF), rectus femoris (RF), biceps femoris (BF), and rectus abdominus (RA). During all trials, sEMG data were sampled at 2000 Hz. Prior to placement, the skin site was prepared by shaving the hair and wiping the skin with alcohol. The skin site preparation and sEMG placement adhered to SENIAM guidelines [47]. All electrodes were placed perpendicular to the muscle belly and on common motor points following current best practices for sEMG placement [47]. The placements were determined by using an anatomical reference chart and through the palpation of the participant by the researchers (JK and MS).

Similar to the relationship between physical activity and muscle activity, a known relationship exists between fat-free mass and relative strength [48]. Ideally, a measure of fat-free mass would have been included in the intake to account for this in the analysis. However, to reduce attrition, this study opted for a brief single-session approach, which rendered a body composition assessment impractical. Instead, total body mass was used and both condition loads (7.2 kg LEO_DB_ or vest) and the loaded hinges (20 kg and 43 kg) were also listed as a percentage of the participant’s total body mass. Spearman’s correlations were then conducted between the percentage of total body mass and each muscle’s activity during each of the trials. As with muscle activity and physical activity, if a moderate (0.4–0.6) to strong (>0.6) correlation5 was found for any of the initial analyses, then that relative load would be used as a control variable in the further analysis of that muscle.

#### 2.4.4. Comfort and Restriction

Upon the completion of the final condition, the participants were given the option to complete a brief questionnaire. The participants rated each of the 4 conditions on a scale from 0 to 10 on comfort and restriction of movement. For a rating of comfort, the participants were asked, “If you had to rank each of the conditions on a scale of 0–10, 10 being the most comfortable ever and 0 being intolerable how would you rate each condition?”. For a rating of restriction, the participants were asked, “If you had to rank each of the conditions on a scale of 0–10, 10 being the most restrictive and 0 being not restrictive at all how would you rate each condition?”. Participants also ranked each of the 4 conditions in reverse order from least restrictive or comfortable (1) to most (4). A prior study assessing tactical load carriage used a similar scale to this study for perceived comfort and restriction of load carriage [49]. Finally, the participants were provided space to give open-ended feedback about their experience wearing the belt during the study protocol.

### 2.5. Statistical Analyses

Following data collection, sEMG data were full-wave-rectified and bandpass-filtered (20 to 490 Hz) with a 4th-order Butterworth filter. The sEMG data were then smoothed using the root mean square computation [50]. Peak muscle activity was then extracted during each trial for each muscle group. The level of sEMG activity for the LEO_DB_ and vest conditions was then normalized to the control condition for the 30 s hinging, 20 kg, and 43 kg lifting trials. Lindner and colleagues previously used a similar normalization procedure to compare load conditions to unloaded [36]. Additionally, this approach supported our primary aim, which was to compare sEMG activity during loaded to unloaded conditions. All sEMG filtering, processing, and calculations were performed in MatLab (MatLab 2020a, MathWorks Inc., Natick, MA, USA).

Variables were initially assessed for normality and extreme values. None of the variables, apart from demographic variables, were found to be normally distributed. Moreover, each dependent variable contained at least one extreme value. Extreme values were defined as a data point that fell greater than 3 standard deviations outside the mean [51]. Thus, to remove these extreme values and minimize data loss, each variable was independently winsorized to the nearest 1% and 99% values [52]. After the removal of the extreme values, normality was reassessed and was still found to be non-normally distributed. Descriptive statistics of the mean and standard deviation were computed.

Prior to analysis, the data were transformed via a Box–Cox transformation, which has been shown to be effective for data sets of small absolute value, which can have a high degree of skewness and heteroscedasticity [53]. However, this transformation did not result in a normal data distribution, nor did other transformations (i.e., log10 and square root). Thus, the sEMG data were analyzed via Friedman tests to assess the effect of condition (4 levels: control, leather LEO_DB_, nylon LEO_DB_, and vest). Post hoc tests were conducted via the Wilcoxon signed rank test when a significant main effect was observed. The effect sizes were calculated with Cliff’s Delta due to the non-normal distribution of the data [54]. Qualitative and quantitative analyses were conducted on the self-reported ratings and ranking of comfort and restriction. A Spearman’s correlation was computed between the ratings of comfort and restriction, as well as the comfort and restriction rankings. Following this, a Friedman test was conducted for each of the variables relative to the control. Any significant model was followed up with a post hoc Wilcoxon signed rank test. Researchers (JWK and MNS) conducted a thematic analysis of the optional participant feedback to determine the themes that were present within the open-ended survey data. The statistical analyses were conducted in R (R-Studios, version 2022.2.0, 4.1.1, R Foundation for Statistical Computing, Vienna, Austria). Statistical significance was set to α ≤ 0.05.

## 3. Results

### 3.1. Physical Activity and Sedentary Behavior

A total of 16 of the 24 included participants were categorized as physically active (≥150 min of physical activity). The participants self-reported a large range of activity levels: MVPA (378.68 ± 466.31 min/week), light physical activity (394.58 ± 653.61 min/week), and sitting time (1016.56 ± 1459.36 min/week). Kendall’s correlations of muscle activity to MVPA, light physical activity, and sitting time only yielded weak (<0.3) to trivial correlations (<0.1). Therefore, physical activity was not accounted for in further analysis.

### 3.2. Muscle Activity

No significant effect for condition was found for any of the eight muscles examined in the three tasks. The results of the Friedman test are displayed in Table 1. A graphical representation of the muscle activity of the three loaded conditions, as percentages of the control, are displayed in Figure 2.

### 3.3. Relative Load

The mean, standard deviation, minimum, and maximum values of relative load for the loaded conditions as well as for the 20 and 43 kg hip hinges can be found in Table 2. Spearman’s correlations of muscle activity to relative load only yielded weak (<0.3) to no present correlation (<0.09). Therefore, relative load was not accounted for in further analysis.

### 3.4. Comfort and Restriction

The results from the participant-rated comfort questionnaire are displayed in Table 3. A significant difference in the rating and rankings of comfort and restriction was found between the loaded conditions and control (*p* < 0.001). Additionally, the vest condition was rated as more comfortable (*p* < 0.05) and less restrictive (*p* < 0.05) than the leather and nylon LEO_DB_. Comfort was significantly correlated with restriction rating (r = 0.791, *p* < 0.05) and ranking (r = −0.848, *p* < 0.05).

From the responses of participants choosing to complete (*n* = 20) the open feedback portion of the exit questionnaire, three general themes emerged. Those themes were as follows: (1) the vest condition was preferred to either LEO_DB_; (2) the LEO_DB_ was uncomfortable, especially the nylon LEO_DB_; and (3) the vest was not preferred to the LEO_DB_. Of those, a majority of responses (*n* = 11) fell into the theme of the LEO_DB_ being uncomfortable, especially the nylon LEO_DB_. The least common response (*n* = 3) was that the vest was not preferred to the LEO_DB_.

Each of the general themes was further categorized into specific subthemes, ranging from two to four responses. Subthemes for the general theme of the LEO_DB_, and especially the nylon LEO_DB_, being uncomfortable included the following: (1) the nylon belt was the least preferred belt due to being the most uncomfortable, (2) the belts were generally uncomfortable and irritating, (3) the belts contributed to a perceived increase in load on the low back and increased low back pressure, and (4) the belts contributed to altered form and comfort during movements. Only one subtheme was present for each of the two remaining general themes. Therefore, these specific subthemes became the remaining general themes of (1) the vest being preferable and (2) the vest not being preferable.

## 4. Discussion

This study aimed to examine whether muscle activity during hip hinge movements was affected by an LEO_DB_ or tactical vest. In support of the hypotheses, no difference in muscle activity was found due to either the LEO_DB_ or tactical vest. However, participants reported that the vest was more comfortable and less restrictive than the LEO_DB_. The lack of effect of load carriage on muscle activity is supported by previous research examining the effects of load carriage on muscle activity during strenuous tasks [55]. For example, it was found in the military population that load carriage did not change muscle activity in the quadratus lumborum, erector spinae, rectus femoris, and gastrocnemius muscles during hiking over rough terrain [55]. However, load carriage has been shown to effect muscle activity in the erector spinae for less strenuous tasks such as incline walking [56] and in the rectus abdominus and multifidus during standing [57]. Therefore, one potential explanation is that the intensity of the physical activity was such that any small changes produced by the 7.2 kg vest or the 7.2 kg LEO_DB_ were masked by the magnitude of the activity from the hinging activities [22,58], meaning that the small sEMG signal change produced by the vest or LEO_DB_ was insignificant compared to the larger signal change produced by the hip hinge task demands.

### 4.1. Muscle Activity and Previous Research

Alternatively, Ouaaid et al. [57] found that as a load was moved inferiorly, muscular activity in the abdominals and lower back was reduced. Thus, due to the proximity of the LEO_DB_ to the center of mass and short moment arm about the hip, the LEO_DB_ might not have been of sufficient magnitude to cause any change based on its location. While this does not explain the findings for the vest condition, it is plausible that the mass of the vest was balanced and dispersed across the trunk in such a way that it too did not lead to any significant change [59]. Lastly, the participants demonstrated a relatively high degree of variability in their responses to the vest and LEO_DB_, and any effect in either direction may have been negated by those who had an inverse response.

### 4.2. Muscle Activity, Physical Activity, and Relative Load

Correlations to muscle activity for both physical activity and relative load were found to be either weak or negligible. High physical activity [60] and relative strength [61,62] have been shown to be associated with better performance in LEOs. Moreover, both physical activity [46] and relative strength [51] are associated with one’s physical health. However, the present study was likely not difficult enough to find an association with these markers. The duration was likely too short to see an impact of physical activity or relative strength on muscle activity brought on by fatigue. In fact, the study was designed in a manner to minimize fatigue to protect the health of the participant. Similarly, the intensity, with respect to load, was likely too low to see shifts in activity and movement mechanics brought on by reaching one’s physical ability. As with fatigue, the participant recruitment was designed in a manner that would exclude individuals for which the loads within the study were near maximum.

This is likely not the case for LEOs during physically demanding tasks, for which these controls are not in place. Therefore, LEO departments and practitioners should not take these findings as an inference that a LEO_DB_ and vest are free from risk. Rather, this initial study found that an LEO_DB_ and vest do not produce quantifiable changes in muscle activity, that are related to LBP, within the first 20 min of wear and while performing hip hinge movements. In other words, an LEO_DB_ and vest are likely not innately harmful for acute wear and the risk they might impose likely stems from factors such as the duration of wear and/or prolonged sitting in an LEO cruiser with an LEO_DB_ or vest.

### 4.3. Comfort and Restriction

Our findings for comfort and restriction with the vest and LEO_DB_ are supported by a growing body of literature on subjective measures for these forms of load carriage. In a study by Schram et al. [30], they found that the larger vest, with a greater desperation of the load across the trunk, was more comfortable than the two smaller vest conditions. This suggests that dispersed loads, such as our vest condition, are perceived as more comfortable than centralized loads. However, body armor, such as a tactical vest, has been shown to be restrictive via reduced measures of mobility, physical performance, and necessary functions such as respiration [33,63]. This may be why some participants reported feeling more restricted in the vest condition versus the belt conditions. In a study by Ramstrand et al. [32], they found that both the vest and LEO_DB_ decreased the measures of range of motion and altered gait patterns. Similar to the present study, Ramstrand et al. reported measures of comfort and perceived LBP and found that most participants preferred the vest to the LEO_DB_ [32]. Thus, the comfort of the load carriage condition might be of greater pragmatic importance than the type of load carriage, for example, accounting for different body shapes and mass distribution such as differences between male and female LEOs. Law enforcement agencies could consider several forms of viable and safe means of load carriage for LEOs and dispense them based on the individual LEO’s preference.

### 4.4. Implications

While this study was specific to the equipment used by LEOs, the findings presented here likely have broader application potential to individuals who wear loads in a similar manner. For example, professions such as carpentry and construction often require wearing a loaded utility belt similar to that of LEOs. Moreover, some individuals choose to carry on their waist with items such as “Fanny Packs”. These items are not identical to the means of load carriage used in the present study. However, seeing as how all these items center load around the waist in a similar manner, some carryover of the present study’s findings is likely. Therefore, individuals who use waist-centered load carriage should not be concerned about it inducing LBP during short-duration, intense activity. Rather, these individuals should consider the functionality of the form of load carriage selected as well as the comfort of wear.

### 4.5. Limitations and Future Recommendations

One delimitation of the current study was the use of absolute loads compared to relative loads. Therefore, the demands of the LEO_DB_ and vest were greater for participants of a smaller stature. This was the case due to the consistency of equipment requirements for LEOs. Today, LEOs are required in most settings to carry a sidearm with magazines, a baton, a radio, pepper spray, hand cuffs, a flashlight, and a taser [64]. Depending on how this equipment is arranged, the weight of the LEO_DB_ can exceed 9 kg. To account for this, a slightly reduced load of 7.2 kg was chosen for this study. However, the standardized load does not consider variations between LEOs and different departments in terms of the content and organization of the LEO_DB_ on an individual basis. Therefore, it is likely that some LEOs experience a greater stress from an LEO_DB_ due to higher load requirements, while other LEOs experience less due to greater dispersion through the use of tools such as thigh holsters [29]. This delimitation does limit the generalizability of these findings to LEO populations and should be considered when interpreting these findings.

A few notable limitations were present within the current study as well. The present findings might not be generalizable to LEOs as we did not control for physical capacities such as muscular strength, muscular endurance, and body composition [64]. Moreover, the use of the IPAQ-SF to assess self-reported physical activity levels did not capture specific details regarding the frequency or experience of participants in performing hip hinging exercises, which may have had an impact on the results. Although not reported, we did initially explore whether body mass was associated with any of the sEMG measures and did not find any significant associations. It has been shown that physical characteristics such as muscular endurance and strength are of particular importance in determining how one responds to load carriage [64,65]. Moreover, since no assessment of strength was conducted, loads present within the study might have been quite high or low relative to the individual’s potential, explaining some of the deviation in responses seen. However, the loads LEOs must carry or move are often absolute in nature. Another limitation is that the present study had participants wear each load condition for approximately 20 min, which would be considered a relatively short duration. Therefore, future research should investigate how the extended wear of an LEO_DB_ effects muscular recruitment and activity. Lastly, to our knowledge, there is not a validated survey instrument to assess the discomfort of load carriage systems, and prior studies reporting the comfort of load carriage used nonvalidated surveys [30,32,49]. The development of a validated survey instrument of perceived load carriage comfort associated with biomechanical measures would be valuable for future research.

### 4.6. Practical Implications

The current study has multiple practical implications for both LEO departments and sports medicine professionals who work with LEOs. First, these findings suggest that a typical LEO_DB_ or tactical vest does not significantly alter muscle activity in a manner that would induce LBP during short-duration, intense activity. Therefore, LEO departments should select equipment that best meets task demands for short-duration, intense activity assignments. Additionally, this evidence suggests that an LEO_DB_ and tactical vest present no additional considerations for sports medicine professionals working with LEOs assigned to highly exertional positions, outside those risks that come with that position assignment. Next, the high variability in these data suggests a high degree of individual difference in regard to both the muscular response to and the comfort and restriction preferences for an LEO_DB_ and tactical vest. LEO departments should consider several appropriate options for their LEOs and allow LEOs to select the equipment based on their personal preference. However, seeing as the LEO_DBs_ and vests used in this study were not field-tested, the primary consideration for LEO departments should remain the functionality of the equipment used. Lastly, sports medicine professionals that commonly work with LEOs should monitor their patients’ varying response to either an LEO_DB_ or tactical vest and adjust protocols based on that individual’s response.

## 5. Conclusions

Overall, evidence on the negative effects of load carriage on muscle activity remains limited. This preliminary study found no effect of an LEO_DB_ and vest on sEMG muscle activity. Future research should investigate the effect of extended LEO_DB_ wear on muscle recruitment, fatigue patterns, and self-reported measures of LBP. Moreover, researchers should investigate if a link between the perceived comfort of load carriage and the future presentation of LBP is present. The present study found that many participants found the vest to be a preferrable form of load carriage compared to the LEO_DB_ in terms of comfort and restriction. Based on this finding, organizations should consider forms of load carriage that meet duty and safety standards as viable and note which method LEOs prefer.

## Figures and Tables

**Figure 1 jfmk-08-00099-f001:**
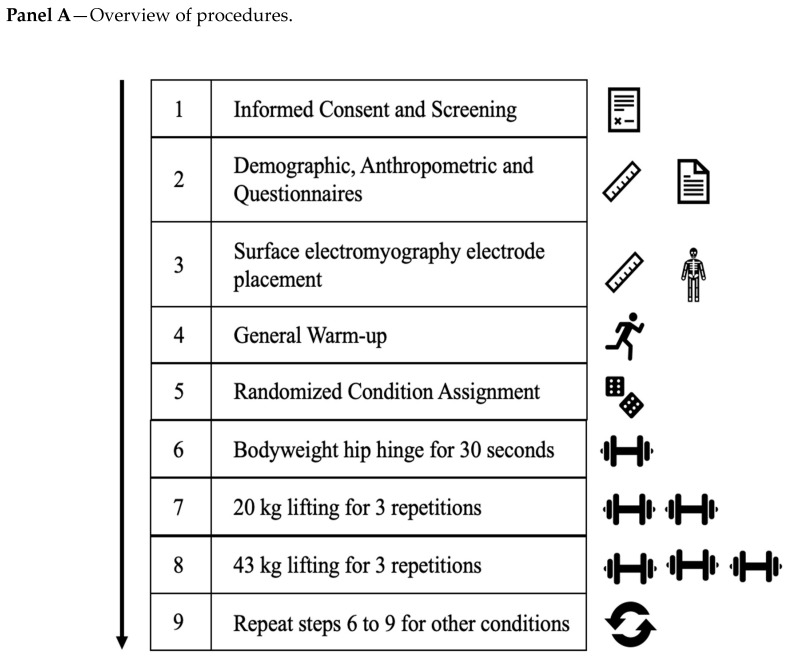
Visual depiction of methods: (**A**) Overview of procedures, (**B**) Visual representation of hip hinge progression.

**Figure 2 jfmk-08-00099-f002:**
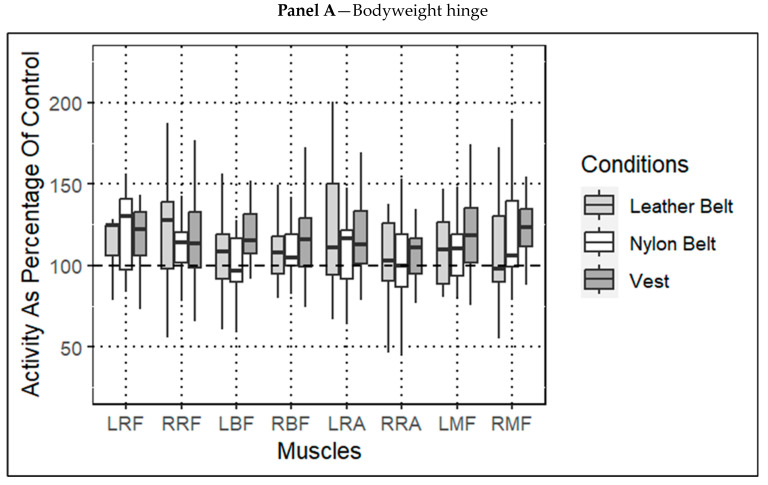
Percentage of muscle activity for each load condition (leather LEODB, nylon LEODB, and vest) compared to control condition for the bodyweight hinge. Dashed line represents values that saw the same activity as control (i.e., 100% of control). Significance was set at *p* < 0.05. Abbreviations: (left rectus femoris = LRF, right rectus femoris = RRF, left biceps femoris = LBF, right biceps femoris = RBF, left rectus abdominis = LRA, right rectus abdominis = RRA, left multifidus = LMF, right multifidus = RMF). Panels: (**A** = sEMG root mean squared (RMS), **B** = peak activity for 20 kg hinge trial, and **C** = peak activity for 43 kg hinge trial). Dashed line indicates 100% of control condition.

**Table 1 jfmk-08-00099-t001:** Results of Friedman test for sEMG data.

	Bodyweight Hinge		20 kg Bar Lift		43 kg Bar Lift	
Muscle	Friedman Test Statistic (X^2^_F_)	*p*-Value	Friedman Test Statistic (X^2^_F_)	*p*-Value	Friedman Test Statistic (X^2^_F_)	*p*-Value
LRF	4.340	0.227	0.671	0.880	1.200	0.753
RRF	0.855	0.836	0.408	0.939	0.328	0.955
LBF	4.810	0.186	0.259	0.967	0.284	0.963
RBF	0.264	0.967	0.278	0.964	0.205	0.977
LRA	1.440	0.695	1.860	0.602	1.800	0.615
RRA	1.180	0.758	0.841	0.840	0.734	0.865
LMF	0.157	0.984	0.798	0.850	0.422	0.936
RMF	0.206	0.977	0.707	0.871	1.020	0.795

*Abbreviations:* (left rectus femoris = LRF, right rectus femoris = RRF, left biceps femoris = LBF, right biceps femoris = RBF, left rectus abdominis = LRA, right rectus abdominis = RRA, left multifidus = LMF, multifidus = RMF).

**Table 2 jfmk-08-00099-t002:** Results from relative load calculation for the loaded conditions and for the two loaded hip hinge trials.

	*Mean*	*STDV*	*Min*	*Max*
** *LEO_DB_ and vest* **	9.67	1.89	6.03	13.85
*20 kg hinge*	26.87	5.25	16.75	38.46
*43 kg hinge*	57.77	11.29	36.01	82.69

Note. The values within this table are the percentage of total body mass.

**Table 3 jfmk-08-00099-t003:** Results from self-reported questionnaire of both restriction and comfort for each condition.

*Comfort*					*Restriction*			
*Condition*	Score		Rank		Score		Rank	
	Mean	STDV	Mean	STDV	Mean	STDV	Mean	STDV
*Control*	-	-	3.96	0.20	-	-	1.00	0.00
*Leather*	5.06	1.80	1.56	0.58	5.04	1.93	3.32	0.69
*Nylon*	5.32	1.82	1.80	0.76	5.72	1.99	3.16	0.75
*Vest*	6.60	2.25	2.68	0.75	7.00	1.94	2.51	0.82

Note. The score column shows values between 0 and 10, with a score of 10 indicating the most comfortable or the least restrictive. The rank column refers to how that condition ranked in terms of comfort or restriction compared to the other conditions. A score of 1 indicates it was the least comfortable or least restrictive and a score of more indicates it was the most comfortable or most restrictive.

## Data Availability

The data and code that support the findings of this study are available from the corresponding author, J.R.M., upon reasonable request.

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
