# Peer review of "Effect of the Law Enforcement Duty Belt on Muscle Activation during Hip Hinging Movements in Young, Healthy Adults"

_jfmk, 2023, doi:10.3390/jfmk8030099_

Round 1

Reviewer 1 Report

It is not easy to review the paper by Kearney and colleagues, since it shows a singular dichotomy. The study has been very well written, and the results are very clear. Nevertheless, there are serious issues that generated some perplexities.

1)     Despite the pre-analysis assumptions, the number of subjects seems to be very small. This problem arises more strongly since the main obtained result (comfort) is mainly of qualitative nature. Increasing the number of subjects may affect this result? Moreover, the brief time (20 minutes) is very different from the hours of pubblic service by LEOs.

2)     Is the difference between the types of belts only related to the material? A more specific description may be needed.

3)     Practical implications should be examined further: comfort is very important, but the usefulness in action is another key factor to hold in account. Are the belts identical from this point of view?

4)     How can these results be translated to obtain implication for general population?

Nevertheless, I choose major revisions, since it’s clear that authors worked with diligence to this paper.

Author Response

Comments to the Author
It is not easy to review the paper by Kearney and colleagues, since it shows a singular dichotomy. The study has been very well written, and the results are very clear. Nevertheless, there are serious issues that generated some perplexities.

Despite the pre-analysis assumptions, the number of subjects seems to be very small. This problem arises more strongly since the main obtained result (comfort) is mainly of qualitative nature. Increasing the number of subjects may affect this result? Moreover, the brief time (20 minutes) is very different from the hours of public service by LEOs.

Response: Thank you for bringing these concerns to our attention. While there is common anecdotal reporting from law enforcement officers that their duty belt causes lower back pain, scientific literature to that effect is still quite sparse. The biomechanics, and specifically the muscle activity, of how the duty belt might contribute to lower back pain is not well understood. We do acknowledge that the duration of wear is a major limitation to this study. The intent with this shorter duration of wear was to see if there were any overt characteristics of the duty belt that could induce lower back pain or acutely alter muscle activity. The findings from the current lead-up study can inform future work that assesses the effects of the duty belt over a longer duration, similar to that of law enforcement officers’ everyday wear.     

In regards to the number of subjects, we estimated the desired sample based on our primary outcome of muscle activity. The qualitative findings related to comfort were a intended as a secondary measure; however, they became more interesting given the lack of differences in muscle activity. Based on other reviews of biomechanical effects of load carriage our sample size may be slightly above the average of previously published studies:

Liew, B.; Morris, S.; Netto, K. The Effect of Backpack Carriage on the Biomechanics of Walking: A Systematic Review and Preliminary Meta-Analysis. J Appl Biomech 2016, 32, 614–629, doi:10.1123/jab.2015-0339.

Martin, J.; Kearney, J.; Nestrowitz, S.; Burke, A.; Sax van der Weyden, M. Effects of Load Carriage on Measures of Postural Sway in Healthy, Young Adults: A Systematic Review and Meta-Analysis. Appl Ergon 2023, 106, 103893, doi:10.1016/j.apergo.2022.103893.

Is the difference between the types of belts only related to the material? A more specific description may be needed.

Response: Thank you for raising this point. It is important to note that the belts were the same outside of the material they were made of. Therefore, on lines 148-149 a sentence was added with this clarification.

Practical implications should be examined further: comfort is very important, but the usefulness in action is another key factor to hold in account. Are the belts identical from this point of view?

Response: That you for highlighting this potential area for misinterpretation. It is critical that our recommendations are not seen in a manner that suggests comfort should supersede functionality. Therefore, a sentence was added on lines 543-545 making this distinction clear.

How can these results be translated to obtain implication for general population?

Response: This is interesting to consider for the present work, thank you for the suggestion. Seeing as many individuals choose to wear loads around their waist, these findings likely do have a broader application. To address this point effectively, a brief paragraph was added from lines 482-492.

Reviewer 2 Report

Dear Authors,

I found this manuscript really interesting and important especially, from the practial point of view. Thus, I have some minor comments to help you to improve this manuscript before publication:

1. Formatting of tables and figures could be improved.

2. I found Table 2 not being necessary - it can be described in text.

Author Response

I found this manuscript really interesting and important especially, from the practical point of view. Thus, I have some minor comments to help you to improve this manuscript before publication:

  1. Formatting of tables and figures could be improved.

Response: Thank you for the positive feedback and for pointing this out to us. We aim to present our data as clearly as possible to the reader. In addition to addressing the point below which likely served as a distraction due to the length of the table. The formatting of the remaining two tables was adjusted to make them easier to read. If further revisions are suggested we are happy to make them and kindly ask the reviewer to  be specific with what can be improved.

  1. I found Table 2 not being necessary - it can be described in text.

Response: Thank you for bringing this to our attention. We do not want to provide unnecessary tables. As such, Table 2 was removed and changed to text in section 3.2. The text version of this table is now from line 386-401.

Reviewer 3 Report

Dear Authors,

You have written an interesting paper focusing on the investigation of the LEODB effect on muscular activity and comparing it to a tactical vest, which is a commonly used alternative to the LEODB.

There are some parts that need to be addressed for greater clarity.

The abstract is well written

Introduction:

The introduction is well-written and uses relevant literature that clearly leads and supports the main study rationale.

Methods:

Thank you for presenting the G*Power calculation for determining sample size and study power. Well done!

Please report if the procedures were according to the Helsinki Declaration.

More info about participants' physical activity is needed as there might be a huge difference if someone is active just 3 times per week but someone 6 times. Report this and their activity as some activities might include more hip hinge movements than others and this might affect your results - or add this in the limitations section.

Please state clearly that you were targeting participants with low back pain - avoid duple negative sentences ( excluded if no... etc) - or am I mistaken? This part is not clear - please rewrite it.

How did you test if they can lift over their body mass - test or a questionnaire/question? report

Line 131 - report the online generator

Report at what time of the day were measurements taken.

Report the break between the tests

You are reporting several tests that were not used in the analysis  - delete them.

It is nice you used the OSHA guidelines, however, if you don't set the weight to be the same according to the % of their body weight the results are misleading. 20 kg weight for a 60kg or 80kg person is a completely different load and body mass/structure plays a vital part in the occupational performance of police and military personnel. The same goes for the 7.2kg belt-vest. This needs to be taken into account in the analysis and discussion of the results. Amend

20kg-hinge / Did you adapt the weights on the floor (add them) for higher participants as they would need to go deeper? report

Overall the discussion does not address the load percentage to body weight which is an important factor and without this generalization of findings is misleading. Therefore please address this issue.

Otherwise a nice flow to the text with a good structure.

Therefore I recommend major revisions.

Kind regards

Minor editing of the English language required

Author Response

You have written an interesting paper focusing on the investigation of the LEODB effect on muscular activity and comparing it to a tactical vest, which is a commonly used alternative to the LEODB.

There are some parts that need to be addressed for greater clarity.

The abstract is well written

Response: Thank you.

Introduction:

The introduction is well-written and uses relevant literature that clearly leads and supports the main study rationale.

Response:  Thank you for the positive comments. We spent substantial time writing and revising the introduction.

Methods:

Thank you for presenting the G*Power calculation for determining sample size and study power. Well done!

Response:  Thank you for all the positive feedback!

Please report if the procedures were according to the Helsinki Declaration.

Response: Thank you for pointing out that this information was not present within the manuscript. All our procedures were in accordance with the Helsinki Declaration and this information has been added at the start of the methods section on lines 106-107.

More info about participants' physical activity is needed as there might be a huge difference if someone is active just 3 times per week but someone 6 times. Report this and their activity as some activities might include more hip hinge movements than others and this might affect your results - or add this in the limitations section.

Response: This is an important consideration worth exploring, so thank you for bringing it up. We ran analyses to check whether there was an influence of PA on our outcome measures and either found weak (0.1-0.3) or trivial (<0.1) correlations between physical activity measures. With this in mind, we did not consider physical activity for further analysis. Information on the physical activity data collection procedures was added to lines 239-258 and the results of the analysis were added to the results section on lines 334-340. The reviewer also makes a good point that being more physically activity would possibly result in more experience with hip hinging movements. In hindsight, a limitation was that we did not collect more screening information on familiarity with hip hinging exercises. As such, we’ve noted this as a limitation on lines 524-526,

Please state clearly that you were targeting participants with low back pain - avoid duple negative sentences ( excluded if no... etc) - or am I mistaken? This part is not clear - please rewrite it.

Response: We apologize for this confusion. No participant with LBP of who has had LBP within the past 6 months was allowed to participate. A sentence was added on line 120-122 to make this point clearer for the reader.

How did you test if they can lift over their body mass - test or a questionnaire/question? Report

Response: Thank you for asking this question. This information was taken via the subjects’ self reporting on the entry questionnaire. While pre-participation testing of the individuals strength would have been ideal, considering the aim of this preliminary work, it was decided that a single testing session would be used to avoid attrition. This information regarding assessment participants deadlift ability was added to line 119.  

Line 131 - report the online generator

Response: Thank you for pointing out this missing piece of information. This study used the Google random number generator and this information has been added to the methods section on line 134-135.

Report at what time of the day were measurements taken.

Response: Thank you for highlighting this oversight as the time of day can have important implications. All testing was done in the afternoon between 12pm and 3pm. This information has been added to the procedures section on lines 163-165.

Report the break between the tests

Response: Thank you for noticing this piece of missing information. Rest is important to ensure participants can give their best effort. Each participant was given 5 minutes of rest between condition trials. This information has been added to the procedures section on lines 163-165.

You are reporting several tests that were not used in the analysis  - delete them.

Response:  Thank you for bringing to our attention that this could be confusing. This information was added in anticipation for questions regarding what other tasks were preferred. However, we can see how this can be confusing for the reader. Therefore, the list of tests performed has been removed from the procedure portion of the methods.

It is nice you used the OSHA guidelines, however, if you don't set the weight to be the same according to the % of their body weight the results are misleading. 20 kg weight for a 60kg or 80kg person is a completely different load and body mass/structure plays a vital part in the occupational performance of police and military personnel. The same goes for the 7.2kg belt-vest. This needs to be taken into account in the analysis and discussion of the results. Amend

Response: The relative versus absolute loads was an issue we discussed heavily during the conceptualization and pilot testing phases of the study. Ultimately we decided to to use absolute loads as most tasks in law enforcement and military are absolute in nature and loads not adjusted to the individuals body size. However, we agree that with the reviewer that this can be addressed better in the manuscript and we see the value this can add to the manuscript. Therefore, extensive revisions were made to effectively address this point. The relative loads for the tactical equipment as well as the hip hinges was correlated to each muscles activity. This was done to see if any relationship was present due to the wide range of participant abilities within the study. No notable correlation were found as noted within the updated manuscript so the research team deemed further analysis unwarranted. We would like to thank you for bringing this to our attention in the aim of producing a stronger manuscript. You can find the changes in the methods section (270-280), results (370-375 and new table 2) and the discussion (442-461).  

20kg-hinge / Did you adapt the weights on the floor (add them) for higher participants as they would need to go deeper? Report

Response: This is a good question and thank you for highlighting the lack of clarity here. The position/height of the load was help constant across all of the hip hinges. While this means that taller participants had to travel a larger range of motion, the start height is similar to the position that all police recruits must maintain for the dummy drag [21]. This point was clarified in the methods section on lines 202-206.

Overall the discussion does not address the load percentage to body weight which is an important factor and without this generalization of findings is misleading. Therefore please address this issue.

Response:  This is an important point worth discussing so thank you for bringing it to our attention. We intentionally kept a constant load for this first study on the duty belt to limit the amount of variables and based on the absolute mass of an actual LEO belt due to required equipment. However, this also does limit the generalizability of our findings considering the loads on the duty belt can vary based on the department and on how the officer positions them. A paragraph in greater detail was added to the manuscript in the limitations section on lines 510-524. 

Round 2

Reviewer 3 Report

Dear Authors

Thank you for addressing all of my questions and suggestions fully. The paper's quality improved. Therefore, in my opinion, the paper is ready to be accepted in its current form.

Kind regards

Minor editing of the English language required